# PaI is getting competitive by training longer

## Abstract

The success of iterative pruning methods in achieving state-of-the-art sparse networks has largely been attributed to improved mask identification and an implicit regularization induced by pruning. We challenge this hypothesis and instead posit that their increased training epochs enable improved optimization. To verify this, we show that pruning at initialization (PaI) is significantly boosted by increased training epochs with repeating (cyclic) learning rate schedules akin to iterative pruning, even outperforming standard iterative pruning methods. The dominant mechanism how this is achieved, as we conjecture, can be attributed to a better exploration of the loss landscape leading to a lower training loss. However, at high sparsity, increased training alone is not enough for competitive performance. A strong coupling between learnt parameter initialization and mask seems to be required. Standard methods obtain this coupling via expensive pruning-training iterations, starting from a dense network. To achieve this with sparse training instead, we propose SCULPT-ing, i.e., cyclic training of any sparse mask followed by a single pruning step to couple the parameters and the mask, which is able to match the performance of state-of-the-art iterative pruning methods in the high sparsity regime at reduced computational cost.

## 1 Introduction

Overparameterization has been a key factor in the tremendous success of deep neural networks across a variety of tasks on vision and language (Bubeck et al., 2023), among others. However, the massive model sizes come with the burden of high computational and memory costs Wu et al. (2022); Luccioni et al. (2023). Hence, to ensure long-term benefits of deep learning for society and climate, it is imperative to improve model efficiency not only at inference time but also during training Kaack et al. (2022).

Neural network sparsification offers a means to reduce the number of parameters of a model while minimally affecting its performance. In addition to computational and memory savings, it can also improve generalization Frankle & Carbin (2019); Paul et al. (2023) and interpretability Chen et al. (2022); Hossain et al. (2024), perform denoising Jin et al. (2022); Wang et al. (2023), and introduce verifiability Narodytska et al. (2020); Albarghouthi (2021). While state-of-the-art iterative pruning methods like Learning Rate Rewinding (LRR) (Renda et al., 2020) or Iterative Magnitude Pruning (IMP) (Frankle & Carbin, 2019) are able to obtain highly performant sparse networks, they require training a dense network over multiple pruning and training iterations, which are computationally demanding.

Instead, pruning at initialization (PaI) methods find a sparse mask at initialization that defines which parameters are pruned i.e. frozen to zero. It thus realizes computational and memory savings from the beginning of model development. While they aim to solve one of our most pressing problems by enabling sparse training from scratch, these methods struggle to keep up with the performance of iterative pruning and often fall short at high sparsities, especially on more complex tasks (Frankle et al., 2021a).

Why is this the case? Recent work has attributed the success of iterative pruning methods to their ability to find better sparse masks (Paul et al., 2023), to train flexibly by enabling more parameter sign flips (Gadhikar & Burkholz, 2024; Zhou et al., 2019), and to identify better trainable parameter initializations of the mask (Frankle et al., 2021a; Kuznedelev et al., 2023).

With the goal to fill the gap between PaI and iterative methods, we investigate to which degree we can transfer successful mechanisms from LRR, regarding its training procedure and mask learning ability, to achieve state-of-the-art performance with PaI methods.

First, we study how LRR achieves peak performance in the sparse regime surpassing the performance of its dense counterpart (see Figure 2(a) and (b)). This typical observation has been attributed to a sparsity induced regularization effect (Frankle & Carbin, 2019; Han et al., 2015; Jin et al., 2022). We offer an alternative explanation. Instead of reaching an optimal sparsity level, we posit that the peak corresponds to longer training with repeated learning rate schedules, finding well generalizing parameters. In the absence of pruning, LRR follows a repeated cyclic training procedure, referred to as cyclic training for the rest of the paper. Such a training procedure also boosts the performance of a dense network above the peak obtained by LRR. While Jin et al. (2022) has also realized that a similar training procedure like LRR, without pruning, could increase the performance of a dense network, they have focused on analyzing the regularization effect of pruning and found that pruning with LRR outperforms a dense network in the presence of label noise.

However, we focus on the optimization benefits of training longer on sparse networks in the absence of label noise. We compare a two learning schedules for training longer, a repeated cyclic training schedule similar to LRR and a one-cycle schedule that has a warmup phase followed by a linear decay for the same number of steps. While both schedules are largely similar in terms of performance, we find that cyclic training is better able to explore the loss landscape and hence choose to conduct our experiments with this cyclic schedule, as it also mimics the LRR training procedure.

We find that dense networks usually outperform pruned networks with our improved cyclic training schedule, highlighting the dominant role cyclic training plays to achieve state-of-the-art performance with LRR. The central insight of our work is, however, that cyclic training substantially boosts the performance of PaI methods like SNIP (Lee et al., 2019) and Synflow (Tanaka et al., 2020) as well as random masks (Liu et al., 2021; Gadhikar et al., 2023) (see Figure 4). Even potential regularization effects of sparsity that mitigate label noise can be realized on sparse masks with cyclic training. These improved PaI masks not only consistently outperform or match LRR in the low sparsity regime, they also achieve state-of-the-art PaI performance in the high sparsity regime in spite of relying on fewer training cycles than LRR. While cyclic PaI can still not compete with LRR at high sparsities, we set out to understand its limiting factors and exploit its merits to enable sparse training even at high sparsity.

In this process, we challenge the assumption that LRR primarily excels at mask learning, as it can accurately measure the importance of trained parameters. Strikingly, we find that cyclic training of a supposedly superior sparse LRR mask with a random initialization does not surpass a cyclically trained random mask (or other PaI masks). As we find, it can still obtain LRR performance (with cyclic training) but only by relying on a parameter initialization that is sufficiently coupled to the mask identification process. Conceptually, this is in line with insights into the lottery ticket hypothesis that suggest, iterative pruning also serves the purpose to identify an initialization that contains information about the task Paul et al. (2023) or at least parameter signs that support retraining Gadhikar & Burkholz (2024). In addition to the findings of Paul et al. (2023), we show uncover insights into iterative pruning methods to find that knowledge of initial parameter signs is sufficient to train a sparse mask with cyclic training and match LRR performance.

These insights suggest that the primary information missing in PaI is the right coupling between mask and parameter initialization. To improve this coupling, we propose SCULPT-ing (**S**parse **C**yclic **U**ti**L**ization of **P**runing and **T**raining), as illlustrated in Figure 1. It starts with a) cyclic training of a (potentially random) sparse mask, which b) is pruned in a single step and c) retrained with a single training cycle. This way, SCULPT-ing transfers the main benefits of iterative pruning, i.e., cyclic training and parameter-mask coupling to sparse training, while requiring fewer computational and memory resources at high sparsity.

Our main contributions are as follows:

- We propose repeated cyclic training as an optimization procedure for sparsely initialized neural networks (including random ones), achieving state-of-the-art pruning at initialization (PaI) performance.

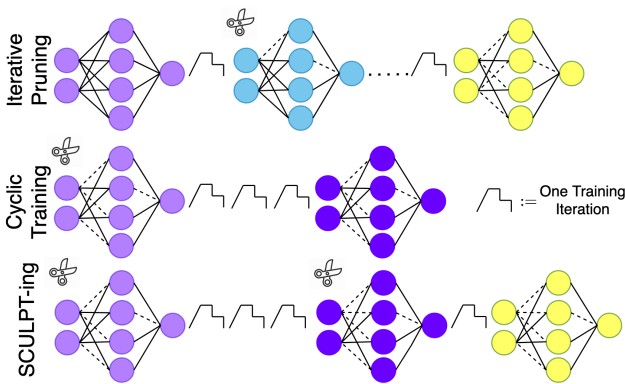

Figure 1: **SCULPT-ing schematic**. A comparison of iterative pruning (top) with cyclic training of a PaI mask (middle) and our proposed method, SCULPT-ing (bottom) to improve PaI.

- At high sparsity, we highlight the importance of an appropriate coupling between parameter initialization and the sparse mask to obtain state-of-the-art performance. In the absence of coupling, we find that the mask learnt by iterative methods induces no benefits over a random mask.
- Based on a rigorous investigation of iterative methods that induce coupling, we propose SCULPT-ing to reach a similar performance as LRR but at reduced computational and memory costs by combining sparse cyclic training with one-shot pruning

## 2 BACKGROUND AND RELATED WORK

**Iterative pruning and lottery tickets.** Iterative pruning methods entail an iterative training and pruning procedure to sparsify neural networks by removing parameters based on an importance measure, usually parameter magnitude. Han et al. (2015) empirically showed the success of these methods on CNNs. Frankle & Carbin (2019) introduced the Lottery Ticket Hypothesis (LTH) and utilized Iterative Magnitude Pruning (IMP) to find sparse, trainable subnetworks of dense randomly initialized source networks, i.e., lottery tickets, that can be trained from scratch to achieve a similar performance as training the dense source network. Although Frankle & Carbin (2019) show the existence of lottery tickets (LTs), they are only able to find them retrospectively by repeating the following steps: a) training a (dense) network, pruning usually 20% of the parameters based on lowest magnitude, c) rewinding the remaining parameters to their initial value. As this approach is less successful on more complex tasks and architectures, Renda et al. (2020) proposed Weight Rewinding (WR) and Learning Rate Rewinding (LRR), which obtain state-of-the-art performance for sparse networks across datasets with iterative pruning. While IMP rewinds to initial weights, WR rewinds to a point obtained after a few training steps, and LRR continues training from the learnt weights of the previous iteration and thus never rewinds the learned neural network parameters. This allows LRR to consistently outperform WR and IMP (Renda et al., 2020; Gadhikar & Burkholz, 2024), yet, we find that repeated cyclic retraining of the WR network is able to fill the gap between WR and LRR.

Zimmer et al. (2023) demonstrate that the retraining duration for LRR can be reduced by adaptively compressing the learning rate schedule based on perturbation induced by pruning. This adaptation while reducing the length of training, does not fully match the performance of LRR, hence, we choose to focus our comparisons to the original LRR method.

**Task specificity of LT initialization.** While the original LTH has given great hope that training sparse neural networks from scratch might be feasible, it has become evident that the mask of sparse LTs Morcos et al. (2019); Chen et al. (2020); Burkholz et al. (2022) as well as the identified initial parameters contain task specific information Paul et al. (2023) that is obtained only by training the dense overparameterized network and it is unclear how to identify mask and initialization otherwise. Theoretical LT existence proofs Malach et al. (2020); Pensia et al. (2020); Orseau et al. (2020); Fischer & Burkholz; Burkolz (2022); Burkholz (2022); da Cunha et al. (2022); Gadhikar et al.

(2023); Ferbach et al. (2023) even suggest that just pruning the random source network can perfectly couple the mask and its initial parameters so that no further training is required. Full task specific information can even be contained in a subset of the parameters Frankle et al. (2021b); Giannou et al. (2023); Burkholz (2024). But is a similar coupling between initialization and mask only attainable by iterative pruning? We propose SCULPT-ing as an alternative.

**Benefits of iterative pruning.** The main motivation of pruning is usually the reduction of computational resources, yet, it has also been found to lead to improved generalization (LeCun et al., 1990; Hassibi et al., 1993; Frankle & Carbin, 2019; Jin et al., 2022) at an optimal sparsity due to repeated training cycles and due to a regularization effect in the presence of label noise. We leverage an improved cyclic training procedure to enable sparse training from scratch. Yet, the success of iterative pruning schemes has been attributed to their ability to transfer crucial information about the loss landscape between consecutive pruning iterations, as they are linearly mode connected (Paul et al., 2023; Du et al.; Frankle et al., 2021a), to find a performant sparse mask and initialization pair. While full training is not necessary to find a good mask (You et al., 2020), training with initial overparameterization in early pruning cycles has been conjectured to improve the mask identification and enable meaningful parameter sign flips during learning (Zhou et al., 2019; Gadhikar & Burkholz, 2024). It is an open question whether PaI could enjoy similar advantages.

**Pruning at initialization (PaI).** PaI methods aim to identify a sparse mask at initialization and enable sparse training from scratch. They use an importance measure like connection sensitivity (SNIP) (Lee et al., 2019), gradient signal preservation (GraSP)(Wang et al., 2020) or criteria that maximize the number of paths while ensuring sufficient widths (Pham et al., 2023; Patil & Dovrolis, 2021; Tanaka et al., 2020) to prune weights. (Liu et al., 2021; Gadhikar et al., 2023) also showed that random pruning is a simple and effective pruning at initialization method which was also earlier verified in sanity checks of mask learning (Su et al., 2020; Ma et al., 2021). We boost their performance significantly with cyclic training.

**Training schedules.** LRR relies on a repeated cyclical learning rate schedule that improves performance at certain sparsities as a consequence of repeated training cycles. Such cyclic training can also improve generalization of dense networks as observed by Jin et al. (2022) and confirmed in Figure 2. Its general benefits for dense training have also been verified by Smith (2017). Defazio et al. (2023) have also highlighted the importance of a linear warmup for improved generalization. Recent work by Kuznedelev et al. (2023) also suggests that sparse networks are under-trained and proposes training them with the AC/DC (Peste et al., 2021) method for increased epochs with a linearly decaying learning rate. Evci et al. (2020) also show that their proposed method RiGL benefits from longer training. However, the success of increased training is attributed to better mask exploration for both methods, which dynamically update the mask during training, and not for a fixed PaI mask which is the focus of our work. Interestingly, we find that simply training longer with cyclic training helps boost the performance of PaI methods.

## 3 EXPERIMENTAL SETUP

We describe the experimental setup used for our investigations. All empirical investigations are performed on image classification tasks, to validate our insights. We train a ResNet20 network for the CIFAR10 Krizhevsky (2009) dataset and use a ResNet18 He et al. (2016) for CIFAR100 and ImageNet Deng et al. (2009) datasets. We also provide additional results on the larger ResNet50 network for CIFAR100 and ImageNet datasets. Our networks were trained on NVIDIA A100 GPUs. All experimental details are provided in Appendix A.2. Accuracy curves in Figures 4, 5,7 and 9 are reported with respect to sparsity i.e. the fraction of zeroed out (pruned) parameters in the network. Sparsity is also given by 1 - density, where density is the fraction of non-zero parameters.

Iterative pruning methods like LRR enjoy the additional benefit of improved generalization performance in comparison with a dense network as shown in Figure 2 Frankle & Carbin (2019); Renda et al. (2020).

**Increased training improves generalization.** Jin et al. (2022) conjectured that parts of this improvement could be attributed to the LRR training schedule but focused their analysis on the additionally induced regularization effect of pruning. To verify that repeated cyclic training benefits generalization, we first train a dense network, without pruning, for the same number of cycles as

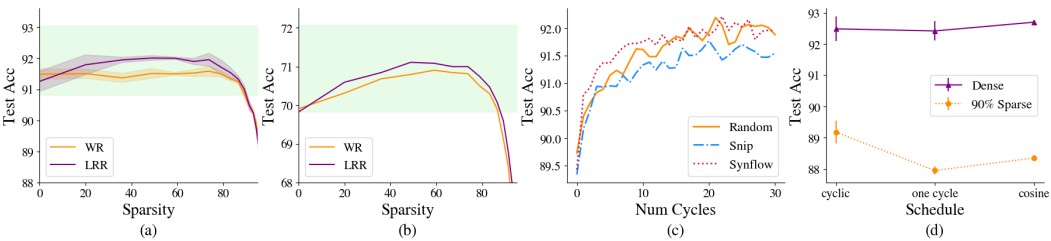

Figure 2: **Increased Cyclic Training**. Comparing LRR and WR for (a) CIFAR10 on ResNet20 and (b) ImageNet on ResNet18. (c) Cyclic training improves performance of a 67% sparse PaI network and (d) Comparing cyclic training to one-cycle training for CIFAR10 on ResNet20.

LRR by repeating the training schedule in each cycle. The green shaded region in Figure 2(a), (b) denotes the improvement in generalization of the dense network with cyclic training over standard training. The dense network sees an increase in performance in the first few cycles, before it plateaus, indicating that only a few additional training cycles are needed to improve the optimization.

**Insights into mechanisms of cyclic training.** Complementary to (Jin et al., 2022), we argue that cyclic training has a strong influence on LRR and also boosts dense training, as suggested by Smith (2017). It also seems to define a generally advantageous learning rate schedule that truly shows its merits in the context of sparse training, which we aim to exploit here. We dedicate this section to investigate the potential mechanisms that could explain its superior performance.

Concretely, we discuss three different but related hypotheses. 1) Training for more epochs is simply better in optimizing the training and test loss. In particular, in the high sparsity regime, Kuznedelev et al. (2023) have encountered that networks tend to be under-trained, in the context of a different pruning method. 2) The regular increase of the learning rate allows cyclic training to effectively jump between local optima in the loss landscape and find flatter optima that have been associated with better generalization Hochreiter & Schmidhuber (1997). 3) Cyclic training finds flatter optima. As it turns out, all three provide a partial explanation, but 2) seems to be the distinguishing factor for which we prefer cyclic training over one-cycle training as discussed below.

## 4 INCREASED (CYCLIC) TRAINING

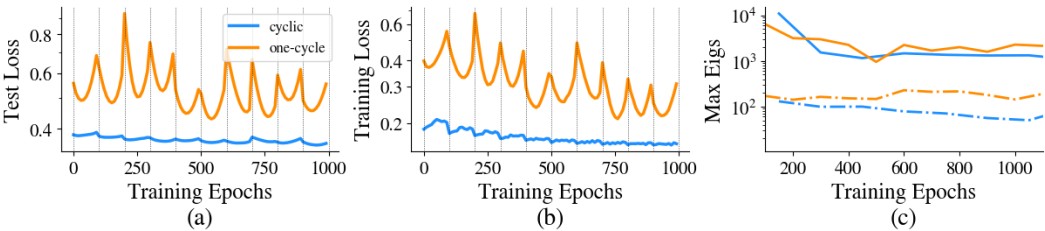

Figure 3: **Insights on cyclic training**. For CIFAR10 on ResNet20: Linear mode connectivity of (a) test and (b) train loss for a 90% sparse random network. (c) Largest eigenvalue of the hessian of the loss function to estimate sharpness. Solid lines denote a 90% sparse network and dashed lines denote a dense network.

**Training longer.** The overall training procedure of LRR takes more training epochs than usual, also because the training cycles have to compensate for pruning operations. Could simply training for longer already improve the generalization performance? To test this hypothesis, we compare cyclic training with two other learning rate schedules, a common one cycle (Defazio et al., 2023) and cosine schedule, which we extend over the same number of training epochs, as visualized in Figure 2(d). Note that the cosine schedule also consists of multiple cycles and thus shares the basic features of cyclic training, yet, the cycle itself is different. According to Figure 2(d), cyclic and cosine schedules are similarly effective in training a dense network and marginally outperform one cycle training, suggesting that the exact LRR schedule might be less special than previously assumed

Jin et al. (2022). Yet, cyclic training is better on a 90% sparse mask, showing promise in the context of sparse training.

**Jumping between local optima.** Repeated cyclic schedules could help escape local minima, in comparison to a single cycle. We confirm this by a linear mode connectivity analysis. Interpolating cyclically trained networks after every cycle and comparing them to interpolated checkpoints for a one-cycle schedule, we observe that consecutive cycles in cyclic training are separated by an error barrier (see Figure 3(b), blue line) in this 1 dimensional training loss landscape. This suggests that cyclic training is able to escape local optima allowing an improved exploration of the loss landscape. On the other hand, one-cycle training has consecutive checkpoints linearly connected (orange line) suggesting they lie in the same optima.

By approximating the largest eigenvalue of the hessian of the loss function as a proxy for flatness, we further confirm in Figure 3(c) that cyclic training ends in a flatter neighborhood, which is known to correlate with improved generalization (Hochreiter & Schmidhuber, 1997; Keskar et al., 2016; Dziugaite & Roy, 2017).

While cyclic as well as one-cycle training seem to share the ability to improve training loss, they are distinguished by the ability of cyclic training to jump optima. Hence, we choose to perform longer training with a cyclic schedule for sparse networks to enable better exploration of the loss landscape.

**Boosting PaI performance with cyclic training.** Having established the benefits of increased (cyclic) training, we propose to exploit it for training sparse masks identified at initialization with PaI. While Kuznedelev et al. (2023) and Evci et al. (2020) have shown that sparse networks benefit from longer training times, they do so with a dynamic mask which can change during training. However, we propose to simply train a fixed mask for longer. *PaI methods which identify a fixed mask, receive a significant boost in performance with increased cyclic training.* Figure 4 shows that, similar to a dense network, cyclic training also improves the generalization of a sparse network, but the boost is larger at higher sparsity. We only need to train the sparse network for enough number of cycles such that the generalization performance peaks, as shown in Figure 2(c).

This enables cyclic PaI to outperform LRR at low sparsity (see also Figure 5). At higher sparsity, cyclic PaI needs fewer training cycles than LRR. Yet, it can only match the performance of LRR on ImageNet at 20% sparsity beyond which the effect of pruning becomes dominant.

**Regularization effect of cyclic PaI.** Complementing the finding by Jin et al. (2022) that LRR pruning increases robustness to label noise, we find that cyclic PaI can realize similar benefits but with initial sparsity according to Figure 9(b)), where a sparse random network generalizes better than a dense one.

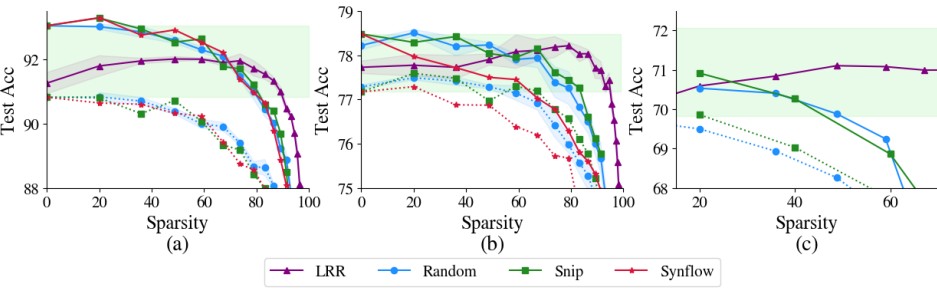

Figure 4: **PaI with cyclic training**. Cyclic training boosts performance of a sparse mask. Shaded region highlights the gain in performance of a dense network by cyclic training for reference. Solid lines denote results with cyclic training and dotted lines show standard training for PaI methods on (a) CIFAR10, ResNet20 (b) CIFAR100, ResNet18 and (c) ImageNet, ResNet18.

**Relevance of the mask.** We observe that different choices of sparse masks using criteria like SNIP or Synflow seem equivalent with cyclic training. Similar conclusions were obtained also in the absence of cyclic training via sanity checks by Liu et al. (2021); Su et al. (2020). However, cyclic PaI is unable to compete with LRR in the high sparsity region. This gap is most pronounced on ImageNet where, although cyclic training improves a random mask considerably, it still falls short compared to LRR. As the optimization procedure for both LRR and a random mask is now similar,

the only difference between them seems to be the sparse mask. However, can we really attribute the gap between LRR and cyclic PaI to task-specific mask learning? As we see in the next section, this conclusion would overlook the central role of the parameter initialization.

**Conclusion.** From this section we conclude that cyclic training can significantly boost PaI methods and even outperform LRR in low sparsity regions, which provides a proof of principle that a strong optimization scheme can make sparse training competitive. The following section seeks to uncover why LRR still performs better in the high sparsity regime.

# 5  DOES THE MASK MATTER?

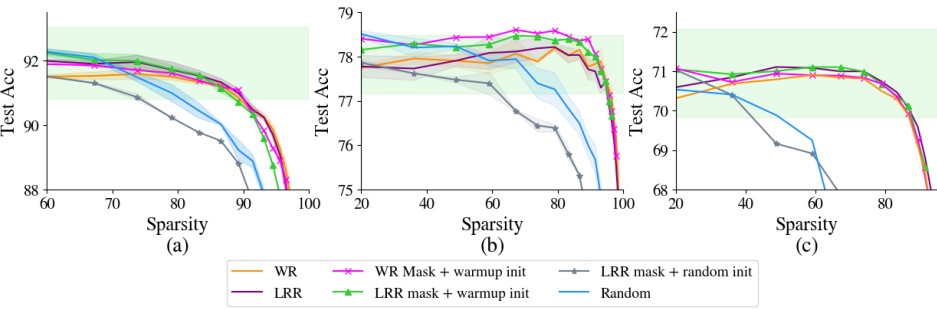

Figure 5: **Coupling**. Comparing cyclic training with combinations of mask and parameter initializations to iterative pruning methods LRR, WR and IMP on (a) CIFAR10, ResNet20 (b) CIFAR100, ResNet18 and (c) ImageNet, ResNet18.

Having established that the learning rate schedule of LRR drives most, but not all of its performance, we are left to wonder what constitutes its strength in the high sparsity regime. The obvious difference between cyclic PaI and LRR are the masks that are optimized. As illustrated in Figure 1, iterative pruning gradually removes the parameters with smallest magnitude in every iteration, thus learning a potentially task specific sparse mask. In contrast, PaI methods identify a sparse mask in a single pruning step at initialization, based on potentially less accurate information.

**LRR learns more than mask structure.** Investigating the sparse mask learnt by LRR, we initialize it with a new random initialization, followed by cyclic training. To our surprise, we observe that the mask identified by LRR with a random initialization is no better than a random mask after cyclic training, as shown in Figure 5 (LRR mask + random init). However, if we initialize the learnt LRR mask with the parameters of a dense network that was trained for a few steps, like in WR, and then perform cyclic training on this combination (LRR mask + warmup init), we are able to recover baseline LRR performance even at high sparsity. This suggests that along with improved optimization via cyclic training, it is crucial to have an appropriate initialization for the sparse mask to improve performance at high sparsity. It also implies that the mask structure learnt by LRR might not be special on its own, but is in combination with the parameter initialization.

**Coupling of parameter initialization and mask.** In order to identify the combinations of parameter initialization and mask that can match LRR at high sparsity, we also look at the masks and initializations of the other iterative pruning methods including WR and optimize each of these mask parameter pairs with cyclic training. We find that, *cyclic training of a WR mask combined with its warmup initialization (WR mask + warmup init) is able to match the performance of LRR*, similar to LRR mask + LRR init. These results, shown in Figure 5, also confirm that when the mask and parameters are coupled, for example as in case of a warmed up initialization and an iteratively learnt mask, they can match the performance of LRR with cyclic training. This insight is particularly interesting, as it suggests that lottery tickets i.e. initializations of sparse networks, might also exist that can achieve LRR performance.

However, is cyclic training really required to achieve this high performance? A linear mode connectivity analysis in Figure 6 further sheds light on the coupling phenomenon. In the case of an LRR mask + random init, consecutive cycles have linearly connected test loss while the training loss has error barriers between cycles. However, for LRR mask + warmup init, we see that consecutive

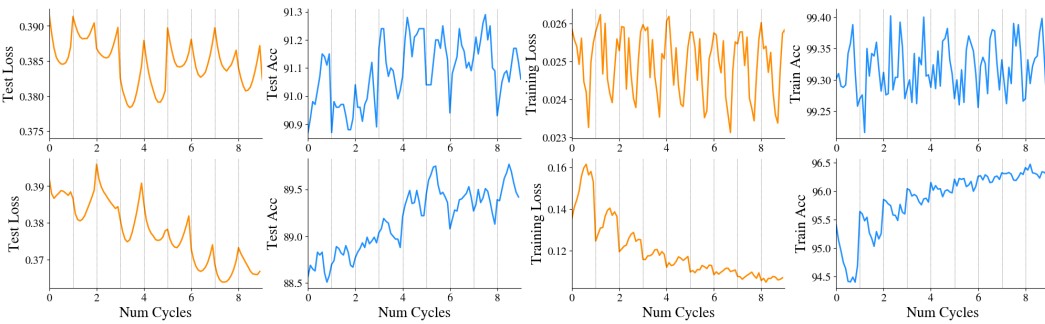

Figure 6: Linear mode connectivity of consecutive training cycles for LRR mask + warmup init (top) and LRR mask + random init (bottom) on CIFAR10, ResNet20 for $90\%$ sparsity.

cycles are mostly in the same loss basin, at least at later stages and enable matching the performance of LRR (see also Figure 18). An initialization that is coupled to the mask and is task specific, starts in the final loss basin or close to it. LRR and WR are known to follow a similarly linearly mode connected optimization trajectory (Paul et al., 2023), while IMP does not enjoy the same benefit, as it always restarts from a random initialization, and struggles to keep up with the performance of LRR and WR, which is in line with our coupling analysis (see also Figure 16).

**Signs are sufficient for coupling.** Based on the findings of Zhou et al. (2019); Gadhikar & Burkholz (2024) who highlight the importance of parameter signs for lottery ticket initializations, we verify if the parameter signs learnt at warmup are already sufficient to couple the mask and parameters. Figure 8(a) shows that using only the parameter signs at warmup with an iteratively pruned mask and random weight magnitudes, cyclic training can bridge the gap between LRR and PaI and even match LRR upto $90\%$ sparsity. This highlights that with the iteratively learnt mask, only warmup signs are sufficient to obtain coupling and improve sparse training, shedding further light on the possibility of lottery ticket initializations.

**Conclusion.** Cyclic training alone is not sufficient to succeed at high sparsity, but requires an initialization that is well coupled to a mask. Our analysis is inconclusive whether LRR masks alone are better aligned with a learning task than PaI masks and poses the potential universality of lottery tickets in the high sparsity regime as an open question Morcos et al. (2019); Chen et al. (2020); Burkholz et al. (2022).

## 6 SCULPT-ING

Our empirical investigations so far have highlighted the potential of cyclic PaI to act as sparse training paradigm, yet, it lacks the right parameter initialization for a given mask and task to compete in the high sparsity regime. Only LRR and to some extent WR have been able to realize the benefits of both the right initialization-mask coupling and cyclic training, as they consistently find highly performant sparse networks. However, both LRR and WR are computationally demanding and memory intensive as they start from a dense network. To enhance sparse training and address the coupling issue, we propose SCULPT-ing, which can achieve a similar performance as LRR and WR while starting sparse network and requiring fewer training cycles at high sparsity. Our experiments verify that SCULPT-ing is often able to bridge the gap between cyclic PaI and LRR at high sparsity.

**SCULPT-ing.** (a) Find a sparse mask at initialization with PaI method of choice. (Our experiments focus on a random mask.) (b) Train with cyclic training to reach peak performance or for the same number of epochs that LRR would take to reach the initial sparsity. (c) Sparsify further by a single step of magnitude pruning to obtain the final sparsity. (d) Retrain with only one training cycle.

The magnitude based pruning step in (c) serves the purpose to couple the learnt parameters to the task and the final sparse mask.

**Experimental results.** SCULPT-ing results are shown in Figure 7. On CIFAR100 with ResNet18, SCULPT $50\%$ matches LRR performance, starting with a $50\%$ sparse random mask. On ImageNet with ResNet18, SCULPT $20\%$ can match LRR starting from a $20\%$ sparse mask, while still being

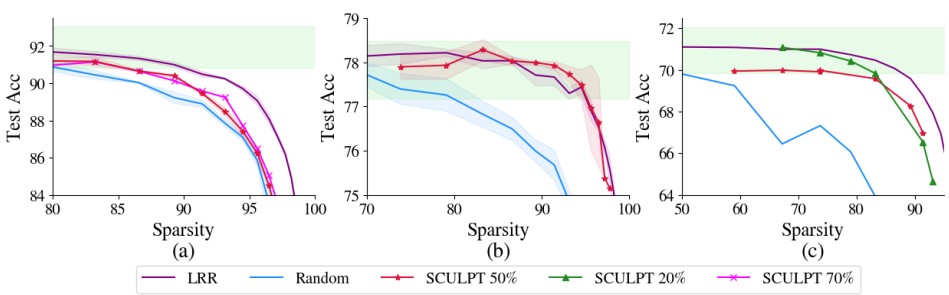

Figure 7: **SCULPT-ing** results starting from a random sparse mask on (a) CIFAR10, ResNet20 (b) CIFAR100, ResNet18 and (c) ImageNet, ResNet18.

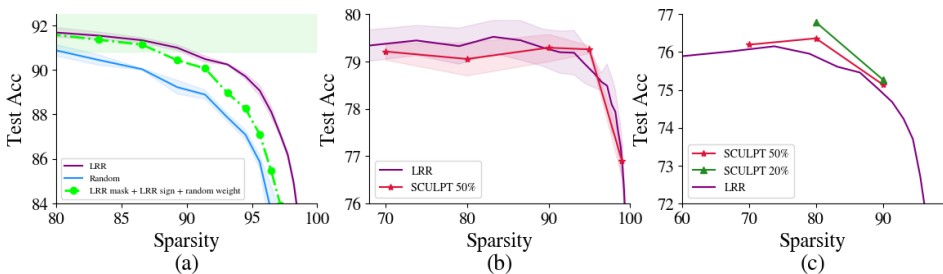

Figure 8: (a) **Signs are sufficient for coupling**, on CIFAR10, ResNet20. **SCULPT-ing** results starting from 50% sparse random mask for CIFAR100, ResNet50 and ImageNet, ResNet50.

competitive if it starts from 50% sparse mask. On CIFAR10 with ResNet20, we can start SCULPT-ing as sparse as 70% and outperform cyclic training of a random mask. However, it is unable to match the performance of LRR. We conjecture that due to the small parameter size of ResNet20, it is less resilient to singleshot pruning with SCULPT-ing.

**SCULPT-ing on larger networks.** We perform SCULPT-ing on the larger ResNet50 network for both CIFAR100 and ImageNet, starting from a 50% sparse random mask. Results in Figure 8(b) and (c). Here, we find that SCULPT-ing is able to match LRR on CIFAR100 and even outperform it on ImageNet. SCULPT-ing benefits from the additional overparametrization of the ResNet50. Moreover, these results establish SCULPT-ing as a viable alternative to LRR.

**Training time.** SCULPT-ing allows sparse networks trained from scratch to compete with and match the performance of LRR and offers two benefits. First, always training a sparse network allows a smaller memory footprint in contrast to LRR which starts from a dense network. Second, the number of training cycles for LRR depends on the final sparsity of the network, i.e., a higher sparsity requires more cycles as every cycle prunes only 20% of nonzero parameters in the original LRR method. SCULPT-ing however uses a flexible number of training cycles for any sparsity and can thus reduce total training cycles at high sparsity. We choose the number of training cycles in SCULPT-ing to maximally boost performance of the sparse mask followed by one additional cycle of retraining after pruning.

For example, in case of a ResNet50 on ImageNet, we start from a 50% sparse random mask and train it for 4 cycles of 90 epochs each, followed by a pruning step for any target sparsity and retraining for 90 epochs in SCULPT-ing. This results in a total of 450 epochs of training for SCULPT. LRR on the other hand requires increasing number of epochs as sparsity increases. At 90% sparsity, LRR needs 10 pruning iterations i.e 900 epochs, whereas SCULPT-ing can achieve similar performance in half the number of epochs. The initial number of training cycles in SCULPT can be traded-off for a smaller boost in performance, to further reduce the training time.

**Magnitude pruning enables coupling.** Figure 9 (a) investigates alternatives to magnitude based pruning in the one-shot pruning step of SCULPT-ing. Interestingly, magnitude seems to be best suitable for realizing a good coupling between mask and its parameters. This might be explained by

the finding that magnitude based pruning minimally changes the neural network function (Mason-Williams & Dahlqvist, 2024). Similarly, second-order pruning methods like Inverse Fisher and WoodFisher could also achieve such coupling (Singh & Alistarh, 2020).

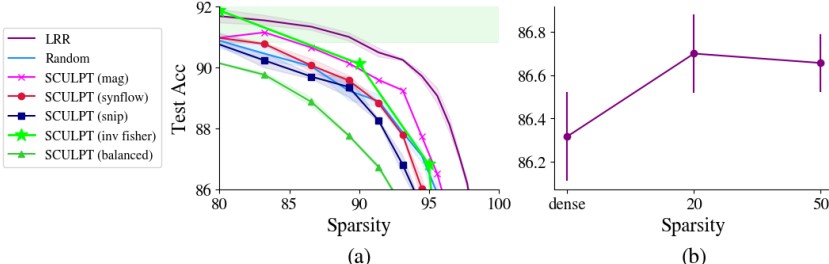

Figure 9: (a) Coupling with different **pruning criteria** with SCULPT-ing on CIFAR10, ResNet20 starting from a 70% sparse random mask. (b) **Regularization effect** of a random sparse mask with cyclic training on CIFAR10, ResNet20 with 15% label noise.

## 7 DISCUSSION

We have conducted a rigorous empirical investigation into the inner mechanisms of state-of-the-art iterative pruning methods Learning Rate Rewinding (LRR) and Weight Rewinding (WR) While their superior performance has largely been attributed to improved mask identification and an implicit sparsity regularization, we have challenged this belief and presented evidence for the insight that their repeated cyclic training schedule enables improved optimization.

To transfer its merits to sparse training, we have proposed to combine cyclic training with pruning at initialization (PaI), which can outperform even LRR at lower sparsity. The performance boost is particularly striking, as Gadhikar & Burkholz (2024) conjectured that mainly early overparameterization supports LRR in learning sparse, highly performant models. As it turns out, a relevant share of its performance and ability to flexibly switch signs is induced by its cyclic training procedure.

Yet, cyclic PaI also faces limits in the high sparsity regime, where we find no significant performance differences between masks, including a mask that has been identified by LRR and can, in principle, achieve a higher performance. This finding identifies a remaining challenge of cyclic PaI, i.e., deriving a parameter initialization that is sufficiently coupled to the mask and learning task so that cyclic training can effectively learn in the high sparsity regime.

To improve this coupling in the context of sparse training, we have proposed SCULPT-ing, which performs cyclic training of a sparse mask followed by a single magnitude based pruning step to induce the desired coupling. SCULPT-ing bridges the gap between sparse training and iterative pruning to save computations in comparison with LRR and improve the performance of cyclic PaI.

While SCULPT-ing can solve a trade-off between computational and performance considerations by adapting its number of training cycles, efficient sparse training remains a challenge that asks for further insights into improved mask identification and effective parameter optimization.

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

# A APPENDIX

## A.1 IMPROVED SIGN RECOVERY BY CYCLIC TRAINING.

Given the importance of sign flips for sparse training (Gadhikar & Burkholz, 2024), we investigate if cyclic training is better at recovering correct signs as compared to one-cycle training for the same number of epochs. We use the coupling experimental setup from Section 5 and train a learnt LRR mask with the signs of the warmup initialization and randomized magnitude with both cyclic and one-cycle training, as reported in Figure 10(b). We find that, with the warmup signs, cyclic training is exactly able to recover LRR performance while one-cycle is worse, while at higher sparsity both cyclic and one-cycle perform identically. Similarly, perturbing 20% of the initial signs in the same also shows that cyclic training can recover better at lower sparsity but is identical to one-cycle at high sparsity.

To further examine the ability of sign recovery, we find that the signs learnt by cyclic training have a 95.37% overlap with the signs learnt by LRR whereas one-cycle has an overlap of 93.67%. A higher overlap with cyclic training suggests that it is better at being able to recover the signs given the signs at warmup.

However, it is also important to note that LRR is also trained cyclicly, which might be the reason why cyclic training of the warmup signs is able to recover the same signs better.

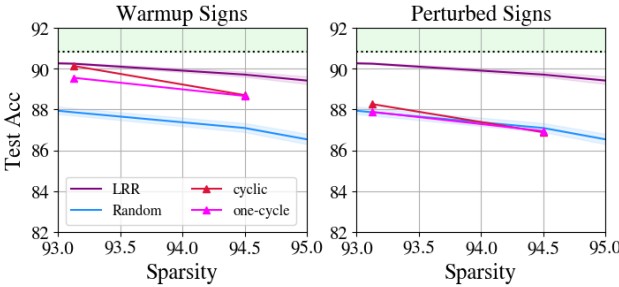

Figure 10: Ability to recover signs for cyclic training in comparison to one-cycle. Results for training a learnt LRR mask with the signs of warmup initialization and random magnitudes (left) and the same with the 20% of the signs also randomly perturbed (right).

We also find that for an LRR mask + warmup init at 93% sparsity, if 20% of the signs are randomly perturbed, cyclic training is able to recover to an accuracy of 88.71% as compared to 88.12% with one-cycle training for 2000 epochs each.

## A.2 EXPERIMENTAL SETUP

The codebase for our experiments was written using PyTorch and torchvision and their relevant primitives for model-construction and data-related operations. In the context of ImageNet experiments we made use of FFCV (Leclerc et al., 2023) for fast dataloading. All models used to report the numbers in in the experiments were trained on a single NVIDIA A100 GPU. We provide all code for our experiments.

We report mean and 95% confidence intervals over 3 seeds for each run in our experiments, except the coupling experiments on CIFAR100 reported and all runs on ImageNet for which we report single runs. All experiments used the SGD optimizer with a weight decay of $1e - 4$ and momentum 0.9. The batch size was fixed to 512 across all experiments and datasets.

When using cyclic training, multiple cycles are used at each sparsity level. Each cycle followed a learning rate schedule as shown in Figure 11(a).

For CIFAR100 and CIFAR10 experiments, each individual training cycle used a multi-step warmup lr scheduler, which starts with a linear-warmup. Each individual cycle has a length of 150 epochs. Subsequent to the warm-up, from an initial learning rate of 0.1, there is reduction by a factor of 10 at epoch 70 and 130. For ImageNet, the cycle length was 90 epochs with a constant warmup for

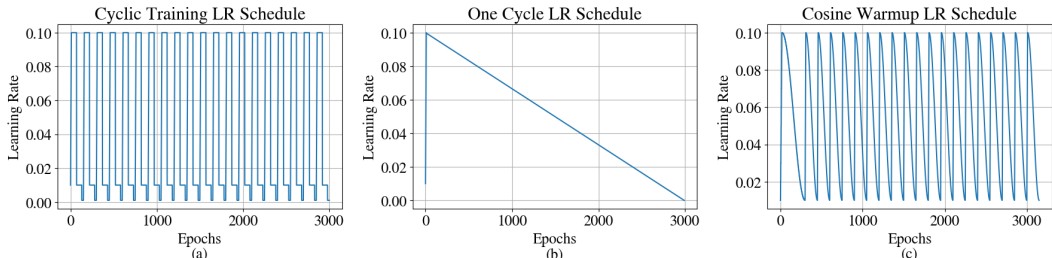

Figure 11: **Left:** The Step Warmup learning rate schedule for a single cycle, initially there is a linear warmup and subsequently there are two steps by a factor of 10 **Middle:** Cyclic Training Learning rate schedule with multiple cycles the schedule in the left plot. **Right:** One Cycle learning rate schedule which uses a fixed cycle over 3000 epochs.

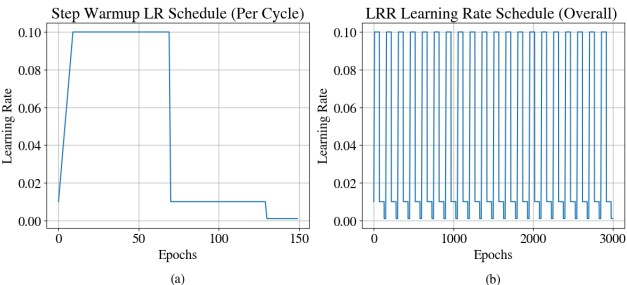

Figure 12: Learning rate schedule when performing standard LRR training across multiple levels. Here we train for one cycle per training level.

10 epochs followed by a step schedule at every 30 epochs with a drop by a factor of 10. For our experiments, we train CIFAR10 and CIFAR100 for upto 8 cycles i.e. 1500 epochs, and 4 cycles i.e. 360 epochs for ImageNet.

In Figure 3, the max eigenvalues were computed using the PyHessian library (Yao et al., 2020).

**Label Noise:** The label noise experiments for CIFAR10 were carried out by randomly flipping 15% using a random permutation of the labels to not impact the balance of labels across the train dataset. The test dataset remains uncorrupted.

## A.3 ITERATIVE MAGNITUDE PRUNING (IMP)

Iterative Magnitude Pruning was introduced by (Frankle & Carbin, 2019) paper. The pruning method can be described as follows:

- Start with an initial dense network $f(x; \theta)$ where $\theta$ drawn from a distribution $D_o$. The objective is to find a mask $m$, to have a network $f(x; m \odot \theta)$ which is sparse.
- This model is then trained as usual, using an algorithm like stochastic gradient descent.
- The parameters of the trained network are then globally ranked according to their magnitude. Then $x\%$ of the lowest valued parameters are set to zero in the mask $m$ which has the exact same size as the network. Typically, $x = 20\%$.
- The parameters that have not been pruned (non-zero) after a pruning level are reset to the initial random initialization $\theta_o$.
- This model is now trained again, repeating steps 2 - 4 until a target sparsity is reached.

## A.4 WEIGHT REWINDING (WR)

One key challenge noticed with IMP in the lottery tickets paper was finding lottery tickets in deeper networks (VGG16 and ResNet). Lottery tickets were found at lower sparsities with use of a learn-

ing rate warmup, but there were none found at higher sparsities. So the authors of (Renda et al., 2020) presented an alternate approach which worked much better. The tickets found are now called "matching tickets".

- Start with an initial dense network $f(x; \theta)$ where $\theta$ drawn from a distribution $D_o$. The objective is to find a mask $m$, to have a network $f(x; m \odot \theta)$ which is sparse.
- The model parameters $\theta_k$ are saved at the $k^{th}$ epoch of dense training (usually after a warmup) that is now used as the rewound initialization.
- This model is then trained as usual, using an algorithm like stochastic gradient descent.
- The parameters of the trained network are then globally ranked according to their magnitude. Then $x\%$ of the lowest valued parameters are set to zero in the mask $m$ which has the exact same size as the network. Typically, $x = 20\%$. This network can be represented by $f(x; m \odot \theta)$
- The parameters that have not been pruned (non-zero) after a pruning level are are now "rewound" to their value in the weight parameters $\theta_k$.
- This model is now trained again, repeating steps 2 - 4 until a target sparsity is reached.

## A.5 LEARNING RATE REWINDING (LRR)

Learning rate rewinding introduced in Renda et al. (2020), instead of resetting/rewinding to the relevant initialization as described above, allows the non-zero parameters to retain their learned values. Instead, LRR at every pruning level resets the learning rate schedule.

- Start with an initial dense network $f(x; \theta)$ where $\theta$ drawn from a distribution $D_o$. The objective is to find a mask $m$, to have a network $f(x; m \odot \theta)$ which is sparse.
- The model parameters $\theta_k$ are saved at the $k^{th}$ epoch of dense training (usually after a warmup) that is now used as the rewound initialization.
- This model is then trained as usual, using an algorithm like stochastic gradient descent.
- The parameters of the trained network are then globally ranked according to their magnitude. Then $x\%$ of the lowest valued parameters are set to zero in the mask $m$ which has the exact same size as the network. Typically, $x = 20\%$. This network can be represented by $f(x; m \odot \theta)$
- This model is now trained again, retaining the learned values of the non-zero weights – repeating steps 2 - 3 until a target sparsity is reached.

| Dataset | CIFAR10 | CIFAR100 | ImageNet |
|---|---|---|---|
| Model | ResNet20 | ResNet18 | ResNet18 |
| Epochs | 150 | 150 | 90 |
| LR | 0.1 | 0.1 | 0.1 |
| Scheduler | step-warmup | step-warmup | step-warmup |
| Batch Size | 512 | 512 | 512 |
| Optimizer | SGD | SGD | SGD |
| Weight Decay | 1e-4 | 1e-3 | 1e-4 |
| Momentum | 0.9 | 0.9 | 0.9 |
| Init | Kaiming Normal | Kaiming Normal | Kaiming Normal |

Table 1: Experimental Setup

## A.6 TRAINING ITERATIONS FOR CYCLIC TRAINING AND LRR.

Figure 2 denotes the total number of training epochs required for LRR and for SCULPT for each sparsity.

| Dataset | LRR | SCULPT |
|---------|-----|--------|
| CIFAR10 | $150 \times$ # iters | 2000 |
| CIFAR100 | $150 \times$ # iters | 2000 |
| ImageNet | $90 \times$ # iters | 540 |

Table 2: Number of training epochs required for LRR vs SCULPT-ing.

### A.7 ERK VS BALANCED SPARSITY RATIOS

We find that balanced layerwise sparsity ratios Gadhikar et al. (2023) find better random sparse masks than ERK sparsity ratios Mocanu et al. (2018) as shown in Figure 13.

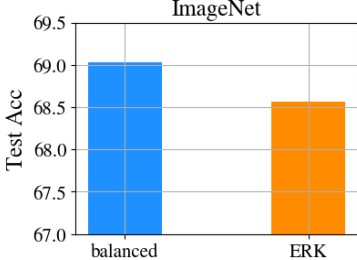

Figure 13: Random masks with different layerwise sparsity ratios on a ResNet18 trained on ImageNet.

### A.8 LINEAR MODE CONNECTIVITY FOR CYCLIC TRAINING.

We provide additional linear mode connectivity plots in support of our claims on the benefits of cyclic training and the importance of coupling.

- Figure 14 shows the connectivity between the first two cycles for cyclic training for a random mask on CIFAR10.
- Figure 15 shows the connectivity between the last two cycles for cyclic training for a random mask on CIFAR10.
- Figure 16 plots the linear mode connectivity of iterative pruning algorithms LRR, WR and IMP as well as an iterative LRR sparse mask with a random init on CIFAR10.
- Figure 17 plots the linear mode connectivity for models every 200 epochs for random sparse networks trained with one-cycle and cosine schedules for 2000 epochs on CIFAR10.
- Figure 18 shows the linear mode connectivity for a LRR mas + warmup init and LRR mask + random init on CIFAR100 at 90% sparsity to highlihgt the phenomenon of coupling.

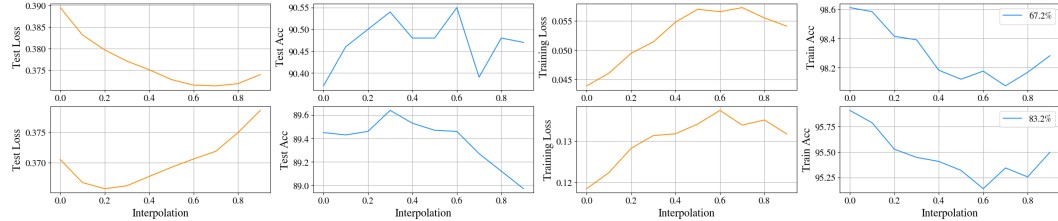

Figure 14: Linear mode connectivity of random networks after standard training i.e. one cycle of training. Each row corresponds to a sparsity.

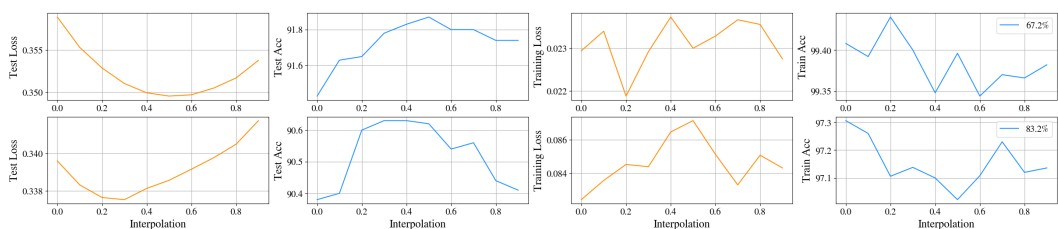

Figure 15: Linear mode connectivity of random networks after repeated cyclic training. Each row corresponds to a sparsity.

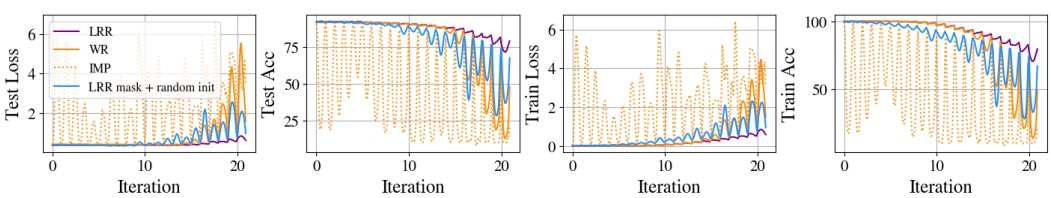

Figure 16: Linear mode connectivity between consecutive masks identified by iterative pruning methods on CIFAR10.

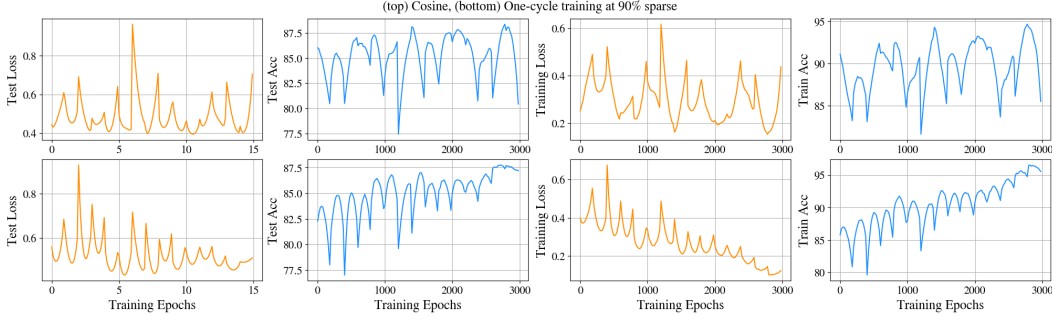

Figure 17: Linear mode connectivity for a 90% sparse random network with increased training using cosine (top) and one-cycle (bottom) learning rate schedules on CIFAR10.

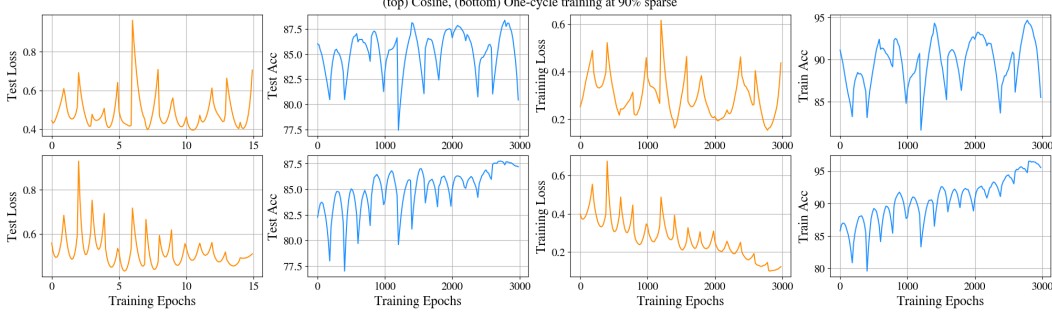

Figure 18: Linear mode connectivity of consecutive training cycles for LRR mask + warmup init (top) and LRR mask + random init (bottom) at 90% sparsity on CIFAR100.

