# OpenReview forum: "PaI is getting competitive by training longer"
_ICLR.cc/2025/Conference — ICLR 2025 Conference Withdrawn Submission_

### Official Review · Reviewer_xyq7 · 2024-10-18

**Soundness:** 1
**Presentation:** 1
**Contribution:** 2
**Rating:** 3
**Confidence:** 3

**Summary:**

This paper studied why pruning at initialization (PaI) was consider being not competive and how to make it competive. It challenged existing hypotheses about iterative pruning methods and showed that training longer, especially with cyclic learning rate schedules, can improve the performance of PaI. Moreover, the importance of coupling between parameter initialization and the sparse mask at high sparsities is reported. The proposed SCULPT-ing method combines cyclic training of a sparse mask and single pruning step to achieve performance comparable to conventional IMP methods with reduced computational cost.

**Strengths:**

- This work focuses on an important topic: why PaI is not competive enough. Novel insights and observations about PaI are presented.

- To my knowledge, the proposed SCULPT-ing method is novel.

- SCULPT-ing is efficient with less memory and computational costs compared with IMP methods.

**Weaknesses:**

-The presentation need to be more clear. For example, SCULPT-ing may be presented more formally as Algorithm 1 with pseudocodes. The bottom line of Figure 1 illustrates that SCULPT-ing requires two cuts, while the discussion some times indicates pruning only happens at initialization.

-The experiments are not comprehensive enough. Only small CNN models are involved. How about Vision/Language Transformers? Note that pruning is more important for those large-scale models, right?

-The work suggests that PaI can be competitive. But how competitive it is exactly? What are the SOTA pruning methods? It will be useful to compare SCULPT-ing with SOTA pruning methods with various model architectures on various benchmark tasks.

-This paper lacks theoretical analysis.

**Questions:**

Please see the weakness.

---

### Official Review · Reviewer_PDzT · 2024-11-03

**Soundness:** 3
**Presentation:** 2
**Contribution:** 1
**Rating:** 3
**Confidence:** 5

**Summary:**

This paper studies the effect of cyclical learning rate schedules and extending training durations on sparse neural networks (SNNs) when initial sparsity is determined using a variety of Pruning-at-Initialization (PaI) methods including random mask initialization. Through a thorough empirical analysis of SNNs trained with a variety of learning rate schedules, training durations, linear mode connectivity, and mask/parameter coupling the authors propose SCULPT, a method which combines PaI with extended training and cyclical learning rates.

**Strengths:**

* Given the ever growing sizes of DNNs, potential methods to improve their efficiency for both training and inference is well motivated. PaI and SNNs in general are one potentially promising avenue to obtain more efficient DNNs.
* The literature review covers most of the relevant literature and sufficiently introduces the key concepts explored in the paper.
* The writing is concise and clear.
* The paper explores the training dynamics of SNNs through a variety of methods such as linear mode connectivity, approximate loss landscape sharpness (as estimated by largest eigenvalue of hessian), and sign-flipping in training of SNNs.
* This work challenges several existing explanations for known phenomena in training of SNNs such dense-to-sparse training methods (such as AC/DC) resulting in better performance due to improved mask topologies instead of improved mask-parameter coupling.

**Weaknesses:**

Fundamentally, I have three major concerns with this work: 1). Lack of novelty for primary contributions and claims, 2). Low model architecture and dataset diversity; and, 3). Low performance of SCULPT relative to existing methods which achieve better generalization with much lower total training FLOPs. Below I expand on these concerns and include actionable requests that, if satisfied, would enable me to raise my initial score.

### Lack of novelty
* Cyclical LR schedulers are commonly employed in training of SNNs and its benefits have been previously established [1-2, 6]
* Similarly, the benefits of extending training durations have been established across a wide range of SNN training methodologies [3-5].
* Based on the above points, I believe the primary contribution of this work is the rigorous evaluation of extended training with cyclical learning rates specifically for the PaI paradigm. Unfortunately, I believe this contribution is of modest significance given its poor performance compared to existing methods as discussed below.

### Model architecture and dataset diversity
* Empirical evidence for the primary claims of this work are motivated through the training of sparse ResNet CNNs on CIFAR-10, CIFAR-100, and Imagenet-1k. Given the prevalence of the transformer architecture, extending these results to a small ViT (DeiT-Tiny for instance) and also exploring alternative CNN architecture such as MobileNet or EfficientNet would improve my confidence that these results would generalize to other modalities and models.
* The primary evidence offered for the benefit of cyclical LR schedules is based on a single ResNet-20 / CIFAR-10 study (Fig 2). In my opinion, this dataset alone is not sufficient to draw strong conclusions. Confirming the benefit of cyclical LR schedules on datasets more indicative of “real-world” data such as ImageNet would improve my confidence in the claims made w.r.t. cyclical LR vs. one-cycle. Given that the rest of the work depends on this result, this point is of critical importance for me to increase my score.

### Performance of SCULPT
* The authors state that SCULPT is a sparse training method; however, SCULPT requires training with very low initial sparsities (20%) to obtain good results on ImageNet. This stands in stark contrast to methods such as RigL and other dynamic sparse training (DST) algorithms that initialize the mask to the final sparsity level (90% for instance) and maintain that sparsity throughout the entire training duration. RigLx5 (~500 epochs) obtains 76.4% with a 90% sparse ResNet-50 on Imagenet vs. SCULPT 20% with 450 epochs obtaining ~75.5% accuracy.
* The above weakness is further exacerbated by methods such as RigL outperforming SCUPLT while maintaining the final target sparsity throughout the entire training process, resulting in a very large decrease in total FLOPs required compared to SCULPT. Even dense-to-sparse methods such as AC/DC likely outperform SCULPT in an iso-flop comparison when accounting for total training flops.
* Further to this, while the authors claim that this initial sparsity level for SCULPT yields potential for memory / computational benefits over LRR, at sparsities such as 20% it is unlikely that any computational benefits can be realized in practice. The most efficient possible condensed representation for a 20% SNNs is to use bitmasks to compress the weights, adding a 1-bit/parameter overhead. Assuming 16 bit floating point weights during training, this means the total memory overhead of a 20% sparse network, in terms of parameter storage, is 86.25% of the dense network ((16 * 0.8 + 1) / 16). Further, at this level of compression, sparse matmul kernels are not efficient so it would be required to scatter the sparse compressed weights back into a sparse, dense tensor, adding some latency overhead as well.
* To better determine the affect of the the initial sparsity, I’d like to see a comparison of SCULPT 20% with GMP* and its accelerated cubic learning rate scheduler.


### Minor concerns / typos
* L404 ‘upto’ -> up to
* LRR description in A.5 should explicitly discuss learning rate rewinding.

[1] L. Yin et al., “Superposing Many Tickets into One: A Performance Booster for Sparse Neural Network Training,” Jun. 21, 2022, arXiv: arXiv:2205.15322. Accessed: Jun. 28, 2022. [Online]. Available: http://arxiv.org/abs/2205.15322

[2] T. Jin, M. Carbin, D. M. Roy, J. Frankle, and G. K. Dziugaite, “Pruning’s Effect on Generalization Through the Lens of Training and Regularization,” presented at the Advances in Neural Information Processing Systems, May 2022. Accessed: Feb. 09, 2024. [Online]. Available: https://openreview.net/forum?id=OrcLKV9sKWp

[3] K. Sreenivasan et al., “Rare Gems: Finding Lottery Tickets at Initialization,” Jun. 02, 2022, arXiv: arXiv:2202.12002. Accessed: Jul. 02, 2022. [Online]. Available: http://arxiv.org/abs/2202.12002

[4] U. Evci, T. Gale, J. Menick, P. S. Castro, and E. Elsen, “Rigging the Lottery: Making All Tickets Winners,” arXiv, arXiv:1911.11134, Jul. 2021. doi: 10.48550/arXiv.1911.11134.

[5] G. Yuan et al., “MEST: Accurate and Fast Memory-Economic Sparse Training Framework on the Edge,” in Advances in Neural Information Processing Systems, Curran Associates, Inc., 2021, pp. 20838–20850. Accessed: Oct. 14, 2022. [Online]. Available: https://proceedings.neurips.cc/paper/2021/hash/ae3f4c649fb55c2ee3ef4d1abdb79ce5-Abstract.html
[6] E. Kurtic and D. Alistarh, “GMP*: Well-Tuned Gradual Magnitude Pruning Can Outperform Most BERT-Pruning Methods,” Dec. 08, 2022, arXiv: arXiv:2210.06384. doi: 10.48550/arXiv.2210.06384.

[7] S. Han, J. Pool, J. Tran, and W. J. Dally, “Learning both Weights and Connections for Efficient Neural Networks,” Oct. 30, 2015, arXiv: arXiv:1506.02626. doi: 10.48550/arXiv.1506.02626.

**Questions:**

* How does SCULPT perform on ViTs and at least one other CNN architecture (MobileNet EfficientNet)?
* Are the benefits of cyclical learning rate as clear when analyzed on ImageNet?
* How does SCULPT compare with RigL, AC/DC, GMP*, and LRR when plotted with an x-axis of total theoretical training FLOPs taking into account sparsity and training durations?

---

### Official Review · Reviewer_b3VF · 2024-11-04

**Soundness:** 3
**Presentation:** 2
**Contribution:** 3
**Rating:** 5
**Confidence:** 4

**Summary:**

The paper examines the effect of cycling learning rate schedules on the performance of Pruning at Initialization methods. The authors find that employing cycling schedules notably enhances the performance of sparse training, regardless of the mask used. However, at high levels of sparsity, this improvement alone does not suffice to match the results of state-of-the-art iterative pruning methods, which achieve a stronger alignment between model parameters and the mask. To address this limitation, the authors introduce SCULPT-ing, a sparse training framework that combines cycling learning with one-step pruning. For certain datasets, this method successfully recovers the performance achieved by Learning Rate Rewinding (LRR).

**Strengths:**

1) The paper establishes a connection between the choice of learning schedules and the performance of sparse training, addressing an intriguing aspect of how training schedules influence sparse training outcomes. The findings indicate that combining cycling training with PaI masks can enhance performance, particularly in lower sparsity scenarios, regardless of the method used to obtain the mask.
2) The proposed SCULPT-ing method, which integrates cycling training and single-shot pruning, offers a straightforward framework that, in certain cases, can match the performance of Learning Rate Rewinding (LRR) in high sparsity settings. This is noteworthy, as the primary distinction in SCULPT-ing compared to cycling training alone is the inclusion of one-shot pruning.

**Weaknesses:**

1) While the general focus on the impact of cycling training on the performance of PaI training is interesting, I find that a significant part of the paper emphasizes the importance of aligning the mask with proper initialization for high sparsity in PaI. This point has already been extensively discussed in prior research on sparse training (e.g., Frankle & Carbin, 2019; Zhou et al., 2019; Chen et al., 2020). What new insights does this paper contribute to this topic? While it is certainly valid to confirm previous findings through empirical studies, I would be cautious about presenting such a validation as one of the main contributions of the paper (as suggested in the second bullet of the "contributions" section). Overall, I appreciate the authors' comprehensive Related Work section, but I am uncertain whether this paper has been effectively positioned within the existing research context in terms of this topic.
2) From an empirical standpoint, the paper focuses exclusively on ResNet models and three datasets: CIFAR-10, CIFAR-100, and ImageNet. Additionally, using ResNets originally designed for ImageNet when training on CIFAR-100 is known to result in overparameterized models, making it relatively straightforward to operate at quite high sparsity levels without significant performance degradation. The authors might consider using CIFAR-specific ResNet architectures, as they do for CIFAR-10, to ensure a more balanced evaluation.
3) Moreover, this focus raises the question of whether the paper’s findings are transferable to models beyond ResNets. While it would be ideal to extend the analysis to architectures such as Transformers (e.g., ViT), even experiments on smaller fully connected networks, or other convolutional architectures (EfficientNet, MobileNet)  could add valuable context and broaden the scope of the study.

**Questions:**

1) In line 213, the statement “Increased training improves generalization” is made, but the experiments in that section primarily demonstrate that cyclic training improves generalization, which is a related but distinct concept. To substantiate the original claim, it would be necessary to evaluate how extending the training duration with the default optimization procedure (or alternative learning rate schedules) affects generalization performance.
2) In Section 5, within the paragraph titled “Coupling of parameter initialization and mask” (starting on line 375), a question is posed about whether cyclic training is essential for achieving strong performance. However, the subsequent discussion only addresses the effect of using random versus rewind masks. It is unclear how this is relevant to answering the original question.
3) Furthermore, while Figure 6 suggests that an LRR mask with random initialization creates distinct training loss plateaus between cycles, Figure 18 does not reflect the same pattern. This indicates that the presence or absence of these loss barriers alone may not account for the performance differences between random initialization and rewinding.
4) Lines 294-295 assert that “the boost is larger at higher sparsity,” but this appears contradictory, as the data seems to indicate a larger improvement at lower sparsities when comparing cyclic training with default training. Plotting the performance difference (delta) could provide more clarity on this point.
5) In the same section, lines 298-299 state, “it can only match [...] ImageNet at 20% sparsity beyond which the effect of pruning becomes dominant.” This statement is ambiguous—both LRR and PaI remove the same proportion of weights, so it’s unclear what specific “effects” of pruning the authors are referring to and why these effects would reduce performance. Clarification on the exact reasons for the performance drop would be helpful.


Overall, some parts of the paper could also benefit from restructuring the layout to improve readability. For example, the section titled “Insights into the mechanism of cyclic training” suggests that guiding principles will be discussed, but instead, it previews the next section and offers three hypotheses for cyclic training's improved performance. In the subsequent section, the number of paragraphs exceeds the number of hypotheses, and some hypotheses are discussed within the same paragraph. Dividing Section 4 into parts that specifically address the three posed questions from the “Insights…” and moving everything else to a separate chapter would make it clearer which claims have been examined and where. Otherwise it is hard to keep track of which questions have been answered, and which are still open.

**Minor Points**
-  In the “Training longer” section (lines 262-263), the authors compare dense and sparse models trained using different learning rate schedules, including a cosine learning rate schedule. If I understand correctly, the period for this schedule is shorter than the total number of iterations, creating a cyclic pattern. Did the authors test the results of cosine annealing up to the minimum learning rate (where no cycles occur and the learning rate decreases iteratively)?
- Typo: In lines 142-143, the steps are labeled “a)” followed by “c)” without a “b).”

In general, due to the limited datasets and models in the evaluation, as well as my concerns about the contribution of the weight coupling insights (see "Weaknesses"), I am slightly more inclined towards a paper that is marginally below the acceptance threshold, but I am open to the discussion during the review period.

---

### Official Review · Reviewer_LC4S · 2024-11-04

**Soundness:** 2
**Presentation:** 2
**Contribution:** 2
**Rating:** 3
**Confidence:** 4

**Summary:**

This paper presents 3 results related to pruning neural networks for vision classification. First, the authors show that cyclic training boosts performance on several methods for pruning at initialization. Second, the paper explores how at random initialization masks obtained from iterative method like learning rate rewinding do not outperform random masks. Finally, the work presents SCULPT-ing, a method that starts with using a pruning at initialization method to prune to low sparsity, performs cyclic training, and then prunes to high sparsity before completing one more cycle of training.

**Strengths:**

I've written the bulk of my review in the following section in order to provide context for my points. I summarize the strengths here:
* Fig. 4 provides the clearest results in the paper, showing the performance gains for PaI methods form training with a cyclic learning rate.
* Fig. 3 explores the difference in loss landscape between cyclic and one cycle training.
* SCULPT-ing does match the performance of LRR for some of the cases shown in Figures 7 and 8.

**Weaknesses:**

Overall, I think the paper requires major revision to clearly frame the results and their relation to prior work. Furthermore, while the paper heads in some interesting directions, I have reservations about the significance of the new results.

When discussing pruning at initialization, there are two questions that have been of general interest:

1. Do there exists masks that, when applied to the network at random initialization, produce a subnetwork that is trainable to the same test accuracy as the original dense network?
2. Can such a mask be found with no or limited training? (The same question is also of interest for the mask applied to the network early in training.)

In essence, to be deemed successful, training with the masks produced by a pruning at initialization method should produce a test accuracy vs. sparsity curve at or above the performance of weight rewinding, e.g. the curves shown in Fig. 2A,B of this paper. These curves have a sparsity level above which the accuracy quickly drops below that of the dense network; before this, they are consistently at or above this accuracy. The reason for interest in these two questions is to drive down the training cost of finding very sparse networks that make no performance compromises compared to training the full network.

**Section 4:** The experiments in Section 4 most straightforwardly engage with these questions. Fig. 4 shows how cyclic training boosts performance for three different methods of pruning the network at initialization: random, SNIP, and Synflow. However, as clearly acknowledged by the paper, the performance falls off with increasing sparsity much earlier than LRR, especially in the case of ResNet-18 on ImageNet where performance falls below the dense network before 50% sparsity. Thus, while cyclic training boosts performance, there is still a significant gap in the key high sparsity regime. (Note: I would reword Line 350: "Having established that the learning rate schedule of LRR drives most, but not all of its performance, we are left to wonder what constitutes strength in the high sparsity regime.": I think this a misleading summary of the results in Section 4. The summary in line 328 is more accurate: "we conclude that cyclic training can significantly boost PaI methods and even outperform LRR in low sparsity regions.")

**Section 5:** Section 5 then explores the relation between the mask and the parameter initialization. In particular, it is shown that starting with the LRR mask from a warmup initialization and then performing cyclic training matches the performance of LRR but not with random initialization; similar results are obtained with the mask from weight rewinding. This is then discussed in terms of coupling of mask and parameter initialization being crucial to improve performance at high sparsity. However, this does not seem to substantially build on previous results. As shown in Fig. 5, WR and LRR have similar performance to begin with, and Appendix D, Fig. 8 of (Paul et al., 2023) shows that the mask obtained from LRR can be retrained from an early rewind point and match the performance of LRR. Thus, the only difference to previous work is cyclic training is used, which I think provides limited additional insight about the coupling of mask and parameter initialization.

The paper does ask in the text whether cyclic training is required to achieve this performance, but no comparisons are made to non-cyclic training with the LRR mask. Rather the loss landscape comparisons in Fig 6 simply compare the warmup init and random init for just cyclic training. Given that the same result held in previous work, my takeaway is that the cyclic training is relatively unimportant here compared to the fact that "an initialization that is coupled to the mask and is task specific starts in the final loss basin or close to it." The two more novel results are: (1) cyclic training with a random mask outperforms the LRR mask at random initialization and (2) the signs of the LRR mask are sufficient with the warmup initialization.

The section concludes with the following:

```
Cyclic training alone is not sufficient to succeed at high sparsity but requires an initialization that is well coupled to a mask. Our analysis is inconclusive whether LRR masks alone are better aligned with a learning task than PaI masks and poses the potential universality of lottery tickets in the high sparsity regime as an open question.
```

In my view, this conclusion has not pushed us forward on either of the two questions for pruning at initialization. For 1, it is saying we need a rewind point that is not random initialization which has been explored at length in multiple papers, including (Frankle et al., 2020, "Linear Mode Connectivity and the Lottery Ticket Hypothesis") and (Paul et al., 2023). And then for 2, it essentially says that other ways of finding a mask at high sparsity other than WR and LRR remains an open question.

**Section 6:** Section 6 then presents SCULPT-ing as a new algorithm that prunes in 2 steps. First the model is pruned to 20%, 50% or 70% sparsity via one of the PaI methods and then cyclic training is performed. Then the network is pruned to the final sparsity by magnitude pruning and one more cycle of training is performed. For many cases, SCULPT-ing still underperforms LRR at high sparsity, and in the case of ResNet-20 on CIFRAR-10, underperforms cyclic training with PaI methods across all sparsities (this is hypothesized to be caused by the small parameter count). Figure 8c shows the method is successful at high sparsities for ResNet-50 trained on ImageNet.

I see SCULPT-ing as more akin to a continual pruning method than PaI. The final mask is not assumed or tested to be effective at random initialization, but rather at the end of a cyclic training procedure. Thus, I would recommend the emphasis of this section be on the any training FLOP win as discussed in the "Training Time" section.

Given this paper is framed around pruning at initialization, several works are missing from a background discussion, including (Sreenivasan et al., NeuRIPS 2022, "Rare Gems: Finding Lottery Tickets at Initialization.") Table 1 of this paper also lists more relevant works that should be discussed.

Editing note: While the green band is Figure 2, 4, 5, 7, 8, and 9 is described in the text, it is never labelled in the figures.

**Questions:**

1. Line 275: "we observe that consecutive cycles in cyclic training are separated by an error barrier (see Figure 3b, blue line)." Are the colors potentially swapped in Figure 3? The orange line appears to have error barriers but the blue line does not.

2. In Figure 5, did you compare the experiments with the LRR mask to non-cyclic training?

2. In Figure 8, is there a reason why ImageNet results not continued out to high sparsity? Wanted to confirm the method continued to match LRR as in the CIFAR-100, ResNet-50 case.

---

### Note · Authors · 2024-11-15

**Comment:**

We would like to thank all the reviewers for providing valuable feedback on our work. We understand that our paper requires major revisions as pointed out by the reviewers with respect to extensive empirical validation and improving the performance of SCULPT-ing as an algorithm. We are grateful to the reviewers for their time and effort.

**Withdrawal Confirmation:**

I have read and agree with the venue's withdrawal policy on behalf of myself and my co-authors.